# Probabilistic Exponential Integrators

**Nathanael Bosch**
Tübingen AI Center, University of Tübingen
`nathanael.bosch@uni-tuebingen.de`

**Philipp Hennig**
Tübingen AI Center, University of Tübingen
`philipp.hennig@uni-tuebingen.de`

**Filip Tronarp**
Lund University
`filip.tronarp@matstat.lu.se`

## Abstract

Probabilistic solvers provide a flexible and efficient framework for simulation, uncertainty quantification, and inference in dynamical systems. However, like standard solvers, they suffer performance penalties for certain *stiff* systems, where small steps are required not for reasons of numerical accuracy but for the sake of stability. This issue is greatly alleviated in semi-linear problems by the *probabilistic exponential integrators* developed in this paper. By including the fast, linear dynamics in the prior, we arrive at a class of probabilistic integrators with favorable properties. Namely, they are proven to be L-stable, and in a certain case reduce to a classic exponential integrator—with the added benefit of providing a probabilistic account of the numerical error. The method is also generalized to arbitrary non-linear systems by imposing piece-wise semi-linearity on the prior via Jacobians of the vector field at the previous estimates, resulting in *probabilistic exponential Rosenbrock methods*. We evaluate the proposed methods on multiple stiff differential equations and demonstrate their improved stability and efficiency over established probabilistic solvers. The present contribution thus expands the range of problems that can be effectively tackled within probabilistic numerics.

## 1 Introduction

Dynamical systems appear throughout science and engineering, and their accurate and efficient simulation is a key component in many scientific problems. There has also been a surge of interest in the intersection with machine learning, both regarding the usage of machine learning methods to model and solve differential equations [36, 18, 35], and in a dynamical systems perspective on machine learning methods themselves [8, 5]. This paper focuses on the numerical simulation of dynamical systems within the framework of *probabilistic numerics*, which treats the numerical solvers themselves as probabilistic inference methods [11, 12, 33]. In particular, we expand the range of problems that can be tackled within this framework and introduce a new class of stable probabilistic numerical methods for stiff ordinary differential equations (ODEs).

Stiff equations are problems for which certain implicit methods perform much better than explicit ones [10]. But implicit methods come with increased computational complexity per step, as they typically require solving a system of equations. *Exponential integrators* are an alternative class of methods for efficiently solving large stiff problems [48, 16, 7, 15]. They are based on the observation that, if the ODE has a semi-linear structure, the linear part can be solved exactly and only the non-linear part needs to be numerically approximated. The resulting methods are formulated in an explicit manner and do not require solving a system of equations, while achieving similar or better stability than implicit methods. However, such methods have not yet been formulated probabilistically.

37th Conference on Neural Information Processing Systems (NeurIPS 2023).

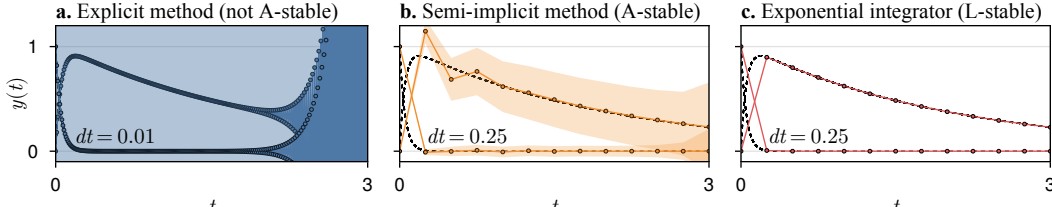

Figure 1: *Probabilistic numerical ODE solvers with different stability properties. Left*: The explicit EK0 solver with a 3-times integrated Wiener process prior is unstable and diverges from the true solution. *Center*: The semi-implicit EK1 with the same prior does not diverge even though it uses a larger step size, due to it being A-stable, but it exhibits oscillations in the initial phase of the solution. *Right*: The proposed exponential integrator is L-stable and thus does not exhibit any oscillations.

In this paper we develop *probabilistic exponential integrators*, a new class of probabilistic numerical solvers for stiff semi-linear ODEs. We build on the *ODE filters* which have emerged as an efficient and flexible class of probabilistic numerical methods for general ODEs [40, 21, 45]. They have known convergence rates [21, 46], which have also been demonstrated empirically [2, 26, 24], they are applicable to a wide range of numerical differential equation problems [23, 25, 3], their probabilistic output can be integrated into larger inference problems [20, 39, 47], and they can be formulated parallel-in-time [4]. But while it has been shown that the choice of underlying Gauss–Markov prior does influence the resulting ODE solver [30, 45, 2], there has not yet been strong evidence for the utility of priors other than the well-established integrated Wiener process. Probabilistic exponential integrators provide this evidence: in the probabilistic numerics framework, "solving the linear part of the ODE exactly" corresponds to an appropriate choice of prior.

**Contributions** Our main contribution is the development of probabilistic exponential integrators, a new class of stable probabilistic solvers for stiff semi-linear ODEs. We demonstrate the close link of these methods to classic exponential integrators in Proposition 1, provide an equivalence result to a classic exponential integrator in Proposition 2, and prove their L-stability in Proposition 3. To enable a numerically stable implementation, we present a quadrature-based approach to directly compute square-roots of the process noise covariance in Section 3.2. Finally, in Section 3.6 we also propose probabilistic exponential Rosenbrock methods for problems in which semi-linearity is not known a priori. We evaluate all proposed methods on multiple stiff problems and demonstrate the improved stability and efficiency of the probabilistic exponential integrators over existing probabilistic solvers.

## 2   Numerical ODE solutions as Bayesian state estimation

Let us first consider an initial value problem with some general non-linear ODE, of the form

$$\dot{y}(t) = f(y(t), t), \qquad t \in [0, T], \tag{1a}$$
$$y(0) = y_0, \tag{1b}$$

with vector field $f : \mathbb{R}^d \times \mathbb{R} \to \mathbb{R}^d$, initial value $y_0 \in \mathbb{R}^d$, and time span $[0, T]$. Probabilistic numerical ODE solvers aim to compute a posterior distribution over the ODE solution $y(t)$ such that it satisfies the ODE on a discrete set of points $\mathbb{T} = \{t_n\}_{n=0}^N \subset [0, T]$, that is

$$p\left(y(t) \,\Big|\, y(0) = y_0, \{\dot{y}(t_n) = f(y(t_n), t_n)\}_{n=0}^N\right). \tag{2}$$

We call this quantity, and approximations thereof, a *probabilistic numerical ODE solution*. Probabilistic numerical ODE solvers thus compute not just a single point estimate of the ODE solution, but a posterior distribution which provides a structured estimate of the numerical approximation error.

In the following, we briefly recapitulate the probabilistic ODE filter framework of Schober et al. [40] and Tronarp et al. [45] and define the prior, data model, and approximate inference scheme. In Section 3 we build on these foundations to derive the proposed probabilistic exponential integrator.

## 2.1 Gauss–Markov prior

*A priori*, we model $y(t)$ with a Gauss–Markov process, defined by a stochastic differential equation

$$\mathrm{d}Y(t) = AY(t)\,\mathrm{d}t + \kappa B\,\mathrm{d}W(t), \qquad Y(0) = Y_0, \tag{3}$$

with state $Y(t) \in \mathbb{R}^{d(q+1)}$, model matrices $A \in \mathbb{R}^{d(q+1) \times d(q+1)}$, $B \in \mathbb{R}^{d(q+1) \times d}$, diffusion scaling $\kappa \in \mathbb{R}$, and smoothness $q \in \mathbb{N}$. More precisely, $A$ and $B$ are chosen such that the state is structured as $Y(t) = [Y^{(0)}(t), \ldots, Y^{(q)}(t)]$, and then $Y^{(i)}(t)$ models the $i$-th derivative of $y(t)$. The initial value $Y_0 \in \mathbb{R}^{d(q+1)}$ must be chosen such that it enforces the initial condition, that is, $Y^{(0)}(0) = y_0$.

One concrete example of such a Gauss–Markov process that is commonly used in the context of probabilistic numerical ODE solvers is the $q$-times Integrated Wiener process, with model matrices

$$A_{\mathrm{IWP}(d,q)} = \begin{bmatrix} 0 & I_d & \cdots & 0 \\ \vdots & \vdots & \ddots & \vdots \\ 0 & 0 & \cdots & I_d \\ 0 & 0 & \cdots & 0 \end{bmatrix}, \qquad B_{\mathrm{IWP}(d,q)} = \begin{bmatrix} 0 \\ \vdots \\ 0 \\ I_d \end{bmatrix}. \tag{4}$$

Alternatives include the class of Matérn processes and the integrated Ornstein–Uhlenbeck process [46]—the latter plays a central role in this paper and will be discussed in detail later.

$Y(t)$ satisfies linear Gaussian transition densities of the form [44]

$$Y(t + h) \mid Y(t) \sim \mathcal{N}\left(\Phi(h)Y(t), \kappa^2 Q(h)\right), \tag{5}$$

with transition matrices $\Phi(h)$ and $Q(h)$ given by

$$\Phi(h) = \exp\left(Ah\right), \qquad Q(h) = \int_0^h \Phi(h - \tau) BB^\top \Phi^\top(h - \tau)\,\mathrm{d}\tau. \tag{6}$$

These quantities can be computed with a matrix fraction decomposition [44]. For $q$-times integrated Wiener process priors, closed-form expressions for the transition matrices are available [21].

## 2.2 Information operator

The likelihood, or data model, of a probabilistic ODE solver relates the uninformed prior to the actual ODE solution of interest with an *information operator* $\mathcal{I}$ [6], defined as

$$\mathcal{I}[Y](t) := E_1 Y(t) - f\left(E_0 Y(t), t\right), \tag{7}$$

where $E_i \in \mathbb{R}^{d \times d(q+1)}$ are selection matrices such that $E_i Y(t) = Y^{(i)}(t)$. $\mathcal{I}[Y]$ then captures how well $Y$ solves the given ODE problem. In particular, $\mathcal{I}$ maps the true ODE solution $y$ to the zero function, i.e. $\mathcal{I}[y] \equiv 0$. Conversely, if $\mathcal{I}[y](t) = 0$ holds for all $t \in [0, T]$ then $y$ solves the given ODE. Unfortunately, it is in general infeasible to solve an ODE exactly and enforce $\mathcal{I}[Y](t) = 0$ everywhere, which is why numerical ODE solvers typically discretize the time interval and take discrete steps. This leads to the data model used in most probabilistic ODE solvers [45]:

$$\mathcal{I}[Y](t_n) = E_1 Y(t_n) - f\left(E_0 Y(t_n), t_n\right) = 0, \qquad t_n \in \mathbb{T} \subset [0, T]. \tag{8}$$

Note that this specific information operator is closely linked to the IVP considered in Eq. (1a). By defining a (slightly) different data model we can also formulate probabilistic numerical IVP solvers for higher-order ODEs or differential-algebraic equations, or encode additional information such as conservation laws or noisy trajectory observations [3, 39].

## 2.3 Approximate Gaussian inference

The resulting inference problem is described by a Gauss–Markov prior and a Dirac likelihood

$$Y(t_{n+1}) \mid Y(t_n) \sim \mathcal{N}\left(\Phi_n Y(t_n), \kappa^2 Q_n\right), \tag{9a}$$

$$Z_n \mid Y(t_n) \sim \delta\left(E_1 Y(t_n) - f\left(E_0 Y(t_n), t_n\right)\right), \tag{9b}$$

with $\Phi_n := \Phi(t_{n+1} - t_n)$, $Q_n := Q(t_{n+1} - t_n)$, initial value $Y(0) = Y_0$, discrete time grid $\{t_n\}_{n=0}^N$, and zero-valued data $Z_n = 0$ for all $n$. The solution of the resulting non-linear Gauss–Markov regression problem can then be efficiently approximated with Bayesian filtering and smoothing techniques [37]. Notable examples that have been used to construct probabilistic numerical ODE solvers include quadrature filters, the unscented Kalman filter, the iterated extended Kalman smoother, or particle filters [19, 45, 46]. Here, we focus on the well-established extended Kalman filter (EKF). We briefly discuss the EKF for the given state estimation problem in the following.

**Prediction**   Given a Gaussian state estimate $Y(t_{n-1}) \sim \mathcal{N}(\mu_{n-1}, \Sigma_{n-1})$ and the linear conditional distribution as in Eq. (9a), the marginal distribution $Y(t_n) \sim \mathcal{N}(\mu_n^-, \Sigma_n^-)$ is also Gaussian, with

$$\mu_n^- = \Phi_{n-1}\mu_{n-1}, \tag{10a}$$

$$\Sigma_n^- = \Phi_{n-1}\Sigma_{n-1}\Phi_{n-1}^\top + \kappa^2 Q_{n-1}. \tag{10b}$$

**Linearization**   To efficiently compute a tractable approximation of the true posterior, the EKF linearizes the information operator $\mathcal{I}$ around the predicted mean $\mu_n^-$, i.e. $\mathcal{I}[Y](t_n) \approx H_n Y(t_n) + b_n$,

$$H_n = E_1 - F_y E_0, \tag{11a}$$

$$b_n = F_y E_0 \mu_n^- - f(E_0\mu_n^-, t_n). \tag{11b}$$

An exact linearization with Jacobian $F_y = \partial_y f(E_0\mu_n^-, t_n)$ leads to a semi-implicit probabilistic ODE solver, which we call the EK1 [45]. Other choices include the zero matrix $F_y = 0$, which results in the explicit EK0 solver [40, 21], or a diagonal Jacobian approximation (the DiagonalEK1) which combines some stability benefits of the EK1 with the lower computational cost of the EK0 [24].

**Correction step**   In the linearized observation model, the posterior distribution of $Y(t_n)$ given the datum $Z_n$ is again Gaussian. Its posterior mean and covariance $(\mu_n, \Sigma_n)$ are given by

$$S_n = H_n \Sigma_n^- H_n^\top, \tag{12a}$$

$$K_n = \Sigma_n^- H_n^\top S_n^{-1}, \tag{12b}$$

$$\mu_n = \mu_n^- - K_n \left( E_1 \mu_n^- - f(E_0\mu_n^-, t_n) \right), \tag{12c}$$

$$\Sigma_n = (I - K_n H_n) \Sigma_n^-. \tag{12d}$$

This is also known as the *update* step of the EKF.

**Smoothing**   To condition the state estimates on all data, the EKF can be followed by a smoothing pass. Starting with $\mu_N^S := \mu_N$ and $\Sigma_N^S := \Sigma_n$, it consists of the following backwards recursion:

$$G_n = \Sigma_n \Phi_n^\top \left( \Sigma_{n+1}^- \right)^{-1}, \tag{13a}$$

$$\mu_n^S = \mu_n + G_n(\mu_{n+1}^S - \mu_{n+1}^-), \tag{13b}$$

$$\Sigma_n^S = \Sigma_n + G_n(\Sigma_{n+1}^S - \Sigma_{n+1}^-)G_n^\top. \tag{13c}$$

**Result**   The above computations result in a *probabilistic numerical ODE solution* with marginals

$$p\left(Y(t_i) \mid \{E_1 Y(t_n) - f(E_0 Y(t_n), t_n) = 0\}_{n=0}^N\right) \approx \mathcal{N}\left(\mu_i^S, \Sigma_i^S\right), \tag{14}$$

which, by construction of the state $Y$, also contains estimates for the ODE solution as $y(t) = E_0 Y(t)$. Since the EKF-based probabilistic solver does not compute only the marginals in Eq. (14), but a full posterior distribution for the continuous object $y(t)$, it can be evaluated for times $t \notin \mathbb{T}$ (also known as "dense output" in the context of ODE solvers); it can produce joint samples from this posterior; and it can be used as a Gauss–Markov prior for subsequent inference tasks [40, 2, 47].

## 2.4   Practical considerations and implementation details

To improve numerical stability and preserve positive-semidefiniteness of the computed covariance matrices, probabilistic ODE solvers typically operate on square-roots of covariance matrices, defined by a matrix decomposition of the form $M = \sqrt{M}\sqrt{M}^\top$ [26]. For example, the Cholesky factor is one possible square-root of a positive definite matrix. But in general, the algorithm does not require the square-roots to be upper- or lower-triangular, or even square. Additionally, we compute the exact initial state $Y_0$ from the IVP using Taylor-mode automatic differentiation [9, 26], we compute smoothing estimates with preconditioning [26], and we calibrate uncertainties globally with a quasi-maximum likelihood approach [45, 2].

# 3 Probabilistic exponential integrators

In the remainder of the paper, unless otherwise stated, we focus on IVPs with a semi-linear vector-field

$$\dot{y}(t) = f(y(t), t) = Ly(t) + N(y(t), t). \tag{15}$$

Assuming $N$ admits a Taylor series expansion around $t$, the variation of constants formula provides a formal expression of the solution at time $t + h$:

$$y(t + h) = \exp(Lh)y(t) + \sum_{k=0}^{\infty} h^{k+1} \left( \int_0^1 \exp(Lh(1 - \tau)) \frac{\tau^k}{k!} \, d\tau \right) \frac{d^k}{dt^k} N(y(t), t). \tag{16}$$

This observation is the starting point for the development of *exponential integrators* [31, 15]. By further defining the so-called $\varphi$-functions

$$\varphi_k(z) = \int_0^1 \exp(z(1 - \tau)) \frac{\tau^{k-1}}{(k-1)!} \, d\tau, \tag{17}$$

the above identity of the ODE solution simplifies to

$$y(t + h) = \exp(Lh)y(t) + \sum_{k=0}^{\infty} h^{k+1} \varphi_{k+1}(Lh) \frac{d^k}{dt^k} N(y(t), t). \tag{18}$$

In this section we develop a class of *probabilistic exponential integrators*. This is achieved by defining an appropriate class of priors that absorbs the partial linearity, which leads to the integrated Ornstein–Uhlenbeck processes. Proposition 1 below directly relates this choice of prior to the classical exponential integrators. Proposition 2 demonstrates a direct equivalence between the predictor-corrector form exponential trapezoidal rule and the once integrated Ornstein–Uhlenbeck process. Furthermore, the favorable stability properties of classical exponential integrators is retained for the probabilistic counterparts as shown in Proposition 3.

## 3.1 The integrated Ornstein–Uhlenbeck process

In Section 2.1 we highlighted the choice of the $q$-times integrated Wiener process prior, which essentially corresponds to modeling the $(q - 1)$-th derivative of the right-hand side $f$ with a Wiener process. Here we follow a similar motivation, but only for the non-linear part $N$. Differentiating both sides of Eq. (15) $q - 1$ times with respect to $t$ yields

$$\frac{d^{q-1}}{dt^{q-1}} \dot{y}(t) = L \frac{d^{q-1}}{dt^{q-1}} y(t) + \frac{d^{q-1}}{dt^{q-1}} N(y(t), t). \tag{19}$$

Then, modeling $\frac{d^{q-1}}{dt^{q-1}} N(y(t), t)$ as a Wiener process and relating the result to $y(t)$ gives

$$dy^{(i)}(t) = y^{(i+1)}(t) \, dt, \tag{20a}$$

$$dy^{(q)}(t) = Ly^{(q)}(t) \, dt + \kappa I_d \, dW^{(q)}(t). \tag{20b}$$

This process is also known as the $q$-times integrated Ornstein–Uhlenbeck process (IOUP), with rate parameter $L$ and diffusion parameter $\kappa$. It can be equivalently stated with the previously introduced notation (Section 2.1), by defining a state $Y(t)$, as the solution of a linear time-invariant (LTI) SDE as in Eq. (3), with system matrices

$$A_{\text{IOUP}(d,q)} = \begin{bmatrix} 0 & I_d & \cdots & 0 \\ \vdots & \vdots & \ddots & \vdots \\ 0 & 0 & \cdots & I_d \\ 0 & 0 & \cdots & L \end{bmatrix}, \qquad B_{\text{IOUP}(d,q)} = \begin{bmatrix} 0 \\ \vdots \\ 0 \\ I_d \end{bmatrix}. \tag{21}$$

*Remark* 1 (The mean of the IOUP process solves the linear part of the ODE exactly). By taking the expectation of Eq. (20b) and by linearity of integration, we can see that the mean of the IOUP satisfies

$$\dot{\mu}^{(0)}(t) = L\mu^{(0)}(t), \qquad \mu^{(0)}(0) = y_0. \tag{22}$$

This is in line with the motivation of exponential integrators: the linear part of the ODE is solved exactly, and we only need to approximate the non-linear part. Figure 2 visualizes this idea.

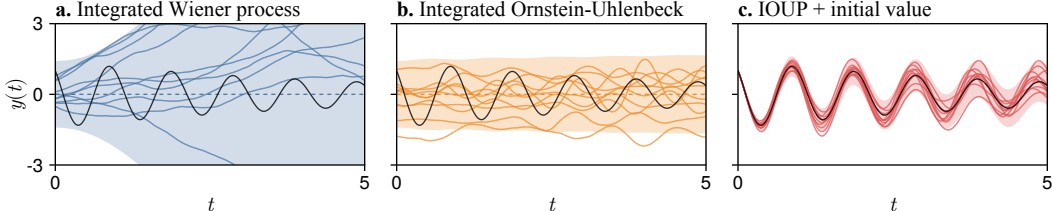

Figure 2: *Damped oscillator dynamics and priors with different degrees of encoded information. Left*: Once-integrated Wiener process, a popular prior for probabilistic ODE solvers. *Center*: Once-integrated Ornstein–Uhlenbeck process (IOUP) with rate parameter chosen to encode the known linearity of the ODE. *Right*: IOUP with both the ODE information and a specified initial value and derivative. This is the kind of prior used in the probabilistic exponential integrator.

## 3.2 The transition parameters of the integrated Ornstein–Uhlenbeck process

Since the process $Y(t)$ is defined as the solution of a linear time-invariant SDE, it satisfies discrete transition densities $p(Y(t+h) \mid Y(t)) = \mathcal{N}\left(\Phi(h)Y(t), \kappa^2 Q(h)\right)$. The following result shows that the transition parameters are intimately connected with the $\varphi$-functions defined in Eq. (17).

**Proposition 1.** *The transition matrix of a $q$-times integrated Ornstein–Uhlenbeck process satisfies*

$$\Phi(h) = \begin{bmatrix} \exp\left(A_{\mathrm{IWP}(d,q-1)}h\right) & \Phi_{12}(h) \\ 0 & \exp(Lh) \end{bmatrix}, \qquad \text{with} \qquad \Phi_{12}(h) := \begin{bmatrix} h^q \varphi_q(Lh) \\ h^{q-1} \varphi_{q-1}(Lh) \\ \vdots \\ h \varphi_1(Lh) \end{bmatrix}. \qquad (23)$$

Proof in Appendix A. Although Proposition 1 indicates that $\Phi(h)$ may be computed more efficiently than by calling a matrix-exponential on a $d(q+1) \times d(q+1)$ matrix, this is known to be numerically less stable [41]. We therefore compute $\Phi(h)$ with the standard matrix-exponential formulation.

**Directly computing square-roots of the process noise covariance**   Numerically stable probabilistic ODE solvers require a square-root, $\sqrt{Q(h)}$, of the process noise covariance rather than the full matrix, $Q(h)$. For IWP priors this can be computed from the closed-form representation of $Q(h)$ via an appropriately preconditioned Cholesky factorization [26]. However, for IOUP priors we have not found an analogous method that works reliably. Therefore, we compute $\sqrt{Q(h)}$ directly with numerical quadrature. More specifically, given a quadrature rule with nodes $\tau_i \in [0, h]$ and positive weights $w_i > 0$, the integral for $Q(h)$ given in Eq. (6) is approximated by

$$Q(h) \approx \sum_{i=1}^{m} w_i \exp(A(h-\tau_i))BB^\top \exp(A^\top(h-\tau_i)) =: \sum_{i=1}^{m} M_i, \qquad (24)$$

with square-roots $\sqrt{M_i} = \sqrt{w_i} \exp(A(h-\tau_i))B$ of the summands, which is well-defined since $w_i > 0$. We can thus compute a square-root representation of the sum with a QR-decomposition

$$X \cdot R = \mathrm{QR}\left(\begin{bmatrix} \sqrt{M_1} & \cdots & \sqrt{M_m} \end{bmatrix}^\top\right). \qquad (25)$$

We obtain $Q(h) \approx R^\top R$, and therefore an approximate square-root factor is given by $\sqrt{Q(h)} \approx R^\top$. Similar ideas have previously been employed for time integration of Riccati equations [42, 43]. We use this quadrature-trick for all IOUP methods, with Gauss–Legendre quadrature on $m = q$ nodes.

## 3.3 Linearization and correction

The information operator of the probabilistic exponential integrator is defined exactly as in Section 2.2. But since we now assume a semi-linear vector-field $f$, we have an additional option for the linearization: instead of choosing the exact $F_y = \partial_y f$ (EK1) or the zero-matrix $F_y = 0$ (EK0), a cheap approximate Jacobian is given by the linear part $F_y = L$. We denote this approach by EKL. This is chosen as the default for the probabilistic exponential integrator. Note that the EKL approach can also be combined with an IWP prior, which will be serve as an additional baseline in the Section 4.

### 3.4 Equivalence to the classic exponential trapezoidal rule in predict-evaluate-correct mode

Now that the probabilistic exponential integrator has been defined, we can establish an equivalence result to a classic exponential integrator, similarly to the closely-related equivalence statement by Schober et al. [40, Proposition 1] for the non-exponential case.

**Proposition 2** (Equivalence to the PEC exponential trapezoidal rule). *The mean estimate of the probabilistic exponential integrator with a once-integrated Ornstein–Uhlenbeck prior with rate parameter $L$ is equivalent to the classic exponential trapezoidal rule in predict-evaluate-correct mode, with the predictor being the exponential Euler method. That is, it is equivalent to the scheme*

$$\tilde{y}_{n+1} = \varphi_0(Lh)y_n + h\varphi_1(Lh)N(\tilde{y}_n), \tag{26a}$$

$$y_{n+1} = \varphi_0(Lh)y_n + h\varphi_1(Lh)N(\tilde{y}_n) + h^2\varphi_2(Lh)\frac{N(\tilde{y}_{n+1}) - N(\tilde{y}_n)}{h}, \tag{26b}$$

*where Eq. (26a) corresponds to a prediction step with the exponential Euler method, and Eq. (26b) corresponds to a correction step with the exponential trapezoidal rule.*

The proof is given in Appendix B. This equivalence result provides another theoretical justification for the proposed probabilistic exponential integrator. But note that the result only holds for the mean, while the probabilistic solver computes additional quantities in order to track the solution uncertainty, namely covariances. These are not provided by a classic exponential integrator.

### 3.5 L-stability of the probabilistic exponential integrator

When solving stiff ODEs, the actual efficiency of a numerical method often depends on its stability. One such property is *A-stability*: It guarantees that the numerical solution of a decaying ODE will also decay, independently of the chosen step size. In contrast, explicit methods typically only decay for sufficiently small steps. In the context of probabilistic ODE solvers, the EK0 is considered to be explicit, but the EK1 with IWP prior has been shown to be A-stable [45]. Here, we show that the probabilistic exponential integrator satisfies the stronger *L-stability*: the numerical solution not only decays, but it decays *fast*, i.e. it goes to zero as the step size goes to infinity. Figure 1 visualizes the different probabilistic solver stabilities. For formal definitions, see for example [27, Section 8.6].

**Proposition 3** (L-stability). *The probabilistic exponential integrator is L-stable.*

The full proof is given in Appendix C. The property essentially follows from Remark 1 which stated that the IOUP solves linear ODEs exactly. This implies fast decay and gives L-stability.

### 3.6 Probabilistic exponential Rosenbrock-type methods

We conclude with a short excursion into exponential Rosenbrock methods [14, 17, 28]: Given a nonlinear ODE $\dot{y}(t) = f(y(t), t)$, exponential Rosenbrock methods perform a continuous linearization of the right-hand side $f$ around the numerical ODE solution and essentially solve a sequence of IVPs

$$\dot{y}(t) = J_n y(t) + (f(y(t), t) - J_n y(t)), \qquad t \in [t_n, t_{n+1}], \tag{27a}$$
$$y(t_n) = y_n, \tag{27b}$$

where $J_n$ is the Jacobian of $f$ at the numerical solution estimate $\hat{y}(t_n)$. This approach enables exponential integrators for problems where the right-hand side $f$ is not semi-linear. Furthermore, by automatically linearizing along the numerical solution the linearization can be more accurate, the Lipschitz-constant of the non-linear remainder becomes smaller, and the resulting solvers can thus be more efficient than their globally linearized counterparts [17].

This can also be done in the probabilistic setting: By linearizing the ODE right-hand side $f$ at each step of the solver around the filtering mean $E_0\mu_n$, we (locally) obtain a semi-linear problem. Then, updating the rate parameter of the integrated Ornstein–Uhlenbeck process at each step of the numerical solver results in *probabilistic exponential Rosenbrock-type methods*. As before, the linearization of the information operator can be done with any of the EK0, EK1, or EKL. But since here the prediction relies on exact local linearization, we will by default also use an exact EK1 linearization. The resulting solver and its stability and efficiency will be evaluated in the following experiments.

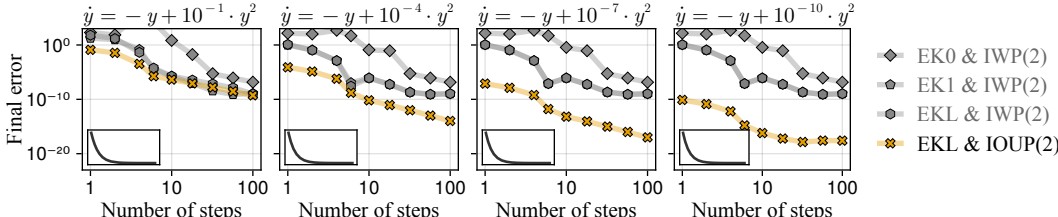

Figure 3: *The IOUP prior is more beneficial with increasing linearity of the ODE.* In all three examples, the IOUP-based exponential integrator achieves lower error while requiring fewer steps than the IWP-based solvers. This effect is more pronounced for the more linear ODEs.

# 4 Experiments

In this section we investigate the utility and performance of the proposed probabilistic exponential integrators and compare it to standard non-exponential probabilistic solvers on multiple ODEs. All methods are implemented in the Julia programming language [1], with special care being taken to implement the solvers in a numerically stable manner, that is, with exact state initialization, preconditioned state transitions, and a square-root implementation [26]. Reference solutions are computed with the DifferentialEquations.jl package [34]. All experiments run on a single, consumer-level CPU. Code for the implementation and experiments is publicly available on GitHub.[1]

## 4.1 Logistic equation with varying degrees of non-linearity

We start with a simple one-dimensional initial value problem: a logistic model with negative growth rate parameter $r = -1$ and carrying capacity $K \in \mathbb{R}_+$, of the form

$$\dot{y}(t) = -y(t) + \frac{1}{K}y(t)^2, \qquad t \in [0, 10], \tag{28a}$$

$$y(0) = 1. \tag{28b}$$

The non-linearity of this problem can be directly controlled through the parameter $K$. Therefore, this test problem lets us investigate the IOUP's capability to leverage known linearity in the ODE.

We compare the proposed exponential integrator to all introduced IWP-based solvers, with different linearization strategies: EK0 approximates $\partial_y f \approx 0$ (and is thus explicit), EKL approximates $\partial_y f \approx -1$, and EK1 linearizes with the correct Jacobian $\partial_y f$. The results for four different values of $K$ are shown in Fig. 3. The explicit solver shows the largest error of all compared solvers, likely due to its lacking stability. On the other hand, the proposed exponential integrator behaves as expected: the IOUP prior is most beneficial for larger values of $K$, and as the non-linearity becomes more pronounced the performance of the IOUP approaches that of the IWP-based solver. Though for large step sizes, the IOUP outperforms the IWP prior even for the most non-linear case with $K = 10$.

## 4.2 Burger's equation

Here, we consider Burger's equation, which is a semi-linear partial differential equation (PDE)

$$\partial_t u(x, t) = D\partial_x^2 u(x, t) - u(x, t)\partial_x u(x, t), \qquad x \in [0, 1], \quad t \in [0, 1], \tag{29}$$

with diffusion coefficient $D \in \mathbb{R}_+$. We transform the problem into a semi-linear ODE with the method of lines [29, 38], and discretize the spatial domain on 250 equidistant points and approximate the differential operators with finite differences. The full IVP specification, including all domains, initial and boundary conditions, and additional information on the discretization, is given in Appendix D.

The results shown in Fig. 4 demonstrate the different stability properties of the solvers: The explicit EK0 with IWP prior is unable to solve the IVP for any of the step sizes due to its insufficient stability, and even the A-stable EK1 and the more approximate EKL require small enough steps $\Delta t < 10^{-1}$. On the other hand, both exponential integrators are able to compute meaningful solutions for a larger range of step sizes. They both achieve lower errors for most settings than their non-exponential

---

[1]https://github.com/nathanaelbosch/probabilistic-exponential-integrators

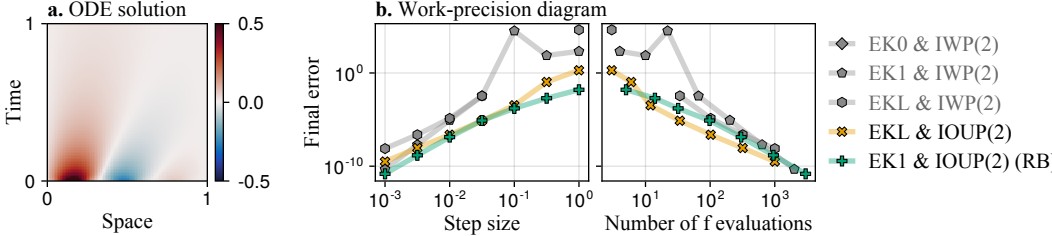

Figure 4: *Benchmarking probabilistic ODE solvers on Burger's equation.* Exponential and non-exponential probabilistic solvers are compared on Burger's equation (a) in two work-precision diagrams (b). Both exponential integrators with IOUP prior achieve lower errors than the existing IWP-based solvers, in particular for large steps. This indicates their stronger stability properties.

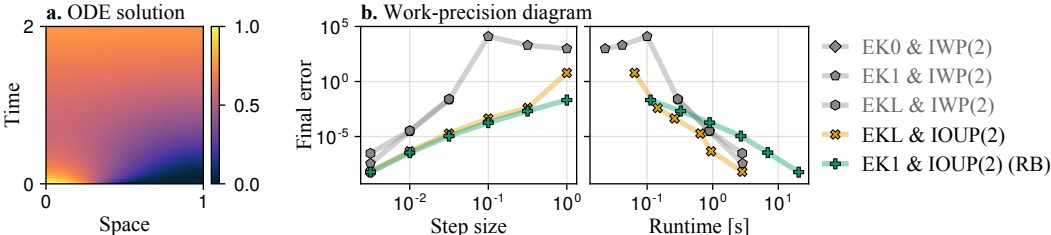

Figure 5: *Benchmarking probabilistic ODE solvers on a reaction-diffusion model.* Exponential and non-exponential probabilistic solvers are compared on a reaction-diffusion model (a) in two work-precision diagrams (b). The proposed exponential integrators with IOUP prior achieve lower errors per step size than the existing IWP-based methods. The runtime comparison shows the increased cost of the Rosenbrock-type (RB) method, while the non-Rosenbrock probabilistic exponential integrator performs best in this comparison.

counterparts. The second diagram in Fig. 4 compares the achieved error to the number of vector-field evaluations and points out a trade-off between both exponential methods: Since the Rosenbrock method additionally computes two Jacobians (with automatic differentiation) per step, it needs to evaluate the vector-field more often than the non-Rosenbrock method. Thus, for expensive-to-evaluate vector fields the standard probabilistic exponential integrator might be preferable.

## 4.3 Reaction-diffusion model

Finally, we consider a discretized reaction-diffusion model given by a semi-linear PDE

$$\partial_t u(x,t) = D\partial_x^2 u(x,t) + R(u(x,t)), \qquad x \in [0,1], \quad t \in [0,T], \tag{30}$$

where $D \in \mathbb{R}_+$ is the diffusion coefficient and $R(u) = u(1-u)$ is a logistic reaction term [22]. A finite-difference discretization of the spatial domain transforms this PDE into an IVP with semi-linear ODE. The full problem specification is provided in Appendix D.

Figure 5 shows the results. We again observe the improved stability of the exponential integrator variants by their lower error for large step sizes, and they outperform the IWP-based methods on all settings. The runtime-evaluation in Fig. 5 also visualizes another drawback of the Rosenbrock-type method: Since the problem is re-linearized at each step, the IOUP also needs to be re-discretized and thus a matrix exponential needs to be computed. In comparison, the non-Rosenbrock method only discretizes the IOUP prior once at the start of the solve. This advantage makes the non-Rosenbrock probabilistic exponential integrator the most performant solver in this experiment.

## 5 Limitations

The probabilistic exponential integrator shares many properties of both classic exponential integrators and of other filtering-based probabilistic solvers. This also brings some challenges.

**Cost of computing matrix exponentials** The IOUP prior is more expensive to discretize than the IWP as it requires computing a matrix exponential. This trade-off is well-known also in the context of classic exponential integrators. One approach to reduce computational cost is to compute the matrix exponential only approximately [32], for example with Krylov-subspace methods [13, 17]. Extending these techniques to the probabilistic solver setting thus poses an interesting direction for future work.

**Cubic scaling in the ODE dimension** The probabilistic exponential integrator shares the complexity most (semi-)implicit ODE solvers: while being linear in the number of time steps, it scales cubically in the ODE dimension. By exploiting structure in the Jacobian and in the prior, some filtering-based ODE solvers have been formulated with linear scaling in the ODE dimension [24]. But this approach does not directly extend to the IOUP-prior. Nevertheless, exploiting known structure could be particularly relevant to construct solvers for specific ODEs, such as certain discretized PDEs.

## 6    Conclusion

We have presented probabilistic exponential integrators, a new class of probabilistic solvers for stiff semi-linear ODEs. By incorporating the fast, linear dynamics directly into the prior of the solver, the method essentially solves the linear part exactly, in a similar manner as classic exponential integrators. We also extended the proposed method to general non-linear systems via iterative re-linearization and presented probabilistic exponential Rosenbrock-type methods. Both methods have been shown both theoretically and empirically to be more stable than their non-exponential probabilistic counterparts. This work further expands the toolbox of probabilistic numerics and opens up new possibilities for accurate and efficient probabilistic simulation and inference in stiff dynamical systems.

## Acknowledgments and Disclosure of Funding

The authors gratefully acknowledge financial support by the German Federal Ministry of Education and Research (BMBF) through Project ADIMEM (FKZ 01IS18052B), and financial support by the European Research Council through ERC StG Action 757275 / PANAMA; the DFG Cluster of Excellence Machine Learning - New Perspectives for Science, EXC 2064/1, project number 390727645; the German Federal Ministry of Education and Research (BMBF) through the Tübingen AI Center (FKZ: 01IS18039A); and funds from the Ministry of Science, Research and Arts of the State of Baden-Württemberg. Filip Tronarp was partially supported by the Wallenberg AI, Autonomous Systems and Software Program (WASP) funded by the Knut and Alice Wallenberg Foundation. The authors thank the International Max Planck Research School for Intelligent Systems (IMPRS-IS) for supporting Nathanael Bosch. The authors also thank Jonathan Schmidt for many valuable discussions and for helpful feedback on the manuscript.

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

# Probabilistic Exponential Integrators — Appendix

## A   Proof of Proposition 1: Structure of the transition matrix

*Proof of Proposition 1.* The drift-matrix $A_{\mathrm{IOUP}(d,q)}$ as given in Eq. (21) has block structure

$$A_{\mathrm{IOUP}(d,q)} = \begin{bmatrix} A_{\mathrm{IWP}(d,q-1)} & E_{q-1} \\ 0 & L \end{bmatrix}, \tag{31}$$

where $E_{q-1} := \begin{bmatrix} 0 & \dots & 0 & I_d \end{bmatrix}^\top \in \mathbb{R}^{dq \times d}$. From Van Loan [48, Theorem 1], it follows

$$\Phi(h) = \begin{bmatrix} \exp\!\big(A_{\mathrm{IWP}(d,q-1)}h\big) & \Phi_{12}(h) \\ 0 & \exp(Lh) \end{bmatrix}, \tag{32}$$

which is precisely Eq. (23). The same theorem also gives $\Phi_{12}(h)$ as

$$\Phi_{12}(h) = \int_0^h \exp(A_{\mathrm{IWP}(d,q-1)}(h-\tau)) E_{q-1}^{(d-1)} \exp(L\tau)\, \mathrm{d}\tau. \tag{33}$$

Its $i$th $d \times d$ block is readily given by

$$\begin{aligned}
(\Phi_{12}(h))_i &= \int_0^h E_i^\top \exp(A_{\mathrm{IWP}(d,q-1)}(h-\tau)) E_{q-1} \exp(L\tau)\, \mathrm{d}\tau \\
&= \int_0^h \frac{(h-\tau)^{q-1-i}}{(q-1-i)!} \exp(L\tau)\, \mathrm{d}\tau \\
&= h^{q-i} \int_0^1 \frac{\tau^{q-1-i}}{(q-1-i)!} \exp(Lh(1-\tau))\, \mathrm{d}\tau \\
&= h^{q-i} \varphi_{q-i}(Lh),
\end{aligned} \tag{34}$$

where the second last equality used the change of variables $\tau = h(1-u)$, and the last line follows by definition. $\qquad\square$

## B   Proof of Proposition 2: Equivalence to a classic exponential integrator

We first briefly recapitulate the probabilistic exponential integrator setup for the case of the once integrated Ornstein–Uhlenbeck process, and then provide some auxiliary results. Then, we prove Proposition 2 in Appendix B.3.

### B.1   The probabilistic exponential integrator with once-integrated Ornstein–Uhlenbeck prior

The integrated Ornstein–Uhlenbeck process prior with rate parameter $L$ results in transition densities $Y(t+h) \mid Y(t) \sim \mathcal{N}\left(Y(t+h); \Phi(h)Y(t), Q(h)\right)$, with transition matrices (from Proposition 1)

$$\Phi(h) = \exp(Ah) = \begin{bmatrix} I & h\varphi_1(Lh) \\ 0 & \varphi_0(Lh) \end{bmatrix}, \tag{35}$$

$$Q(h) = \int_0^h \exp(A\tau) BB^\top \exp(A^\top \tau)\, \mathrm{d}\tau \tag{36}$$

$$= \int_0^h \begin{bmatrix} I & \tau\varphi_1(L\tau) \\ 0 & \varphi_0(L\tau) \end{bmatrix} \begin{bmatrix} 0 & 0 \\ 0 & I \end{bmatrix} \begin{bmatrix} I & \tau\varphi_1(L\tau) \\ 0 & \varphi_0(L\tau) \end{bmatrix}^\top \mathrm{d}\tau \tag{37}$$

$$= \int_0^h \begin{bmatrix} \tau^2\varphi_1(L\tau)\varphi_1(L\tau)^\top & \tau\varphi_1(L\tau)\varphi_0(L\tau)^\top \\ \tau\varphi_0(L\tau)\varphi_1(L\tau)^\top & \varphi_0(L\tau)\varphi_0(L\tau)^\top \end{bmatrix} \mathrm{d}\tau, \tag{38}$$

where we assume a unit diffusion $\sigma^2 = 1$. To simplify notation, we assume an equidistant time grid $\mathbb{T} = \{t_n\}_{n=0}^N$ with $t_n = n \cdot h$ for some step size $h$, and we denote the constant transition matrices simply by $\Phi$ and $Q$ and write $Y_n = Y(t_n)$.

Before getting to the actual proof, let us also briefly recapitulate the filtering formulas that are computed at each solver step. Given a Gaussian distribution $Y_n \sim \mathcal{N}(Y_n; \mu_n, \Sigma_n)$, the prediction step computes

$$\mu_{n+1}^- = \Phi \mu_n, \tag{39}$$

$$\Sigma_{n+1}^- = \Phi(h) \Sigma_n \Phi(h)^\top + Q(h). \tag{40}$$

Then, the combined linearization and correction step compute

$$\hat{z}_{n+1} = E_1 \mu_{n+1}^- - f(E_0 \mu_{n+1}^-), \tag{41}$$

$$S_{n+1} = H \Sigma_{n+1}^- H^\top, \tag{42}$$

$$K_{n+1} = \Sigma_{n+1}^- H^\top S_{n+1}^{-1}, \tag{43}$$

$$\mu_{n+1} = \mu_{n+1}^- - K_{n+1} \hat{z}_{n+1}, \tag{44}$$

$$\Sigma_{n+1} = \Sigma_{n+1}^- - K_{n+1} S_{n+1} K_{n+1}^\top, \tag{45}$$

with observation matrix $H = E_1 - LE_0 = [-L \quad I]$, since we perform the proposed EKL linearization.

## B.2 Auxiliary results

In the following, we show some properties of the transition matrices and the covariances that will be needed in the proof of Proposition 2 later.

First, note that by defining $\varphi_0(z) = \exp z$, the $\varphi$-functions satisfy the following recurrence formula:

$$z\varphi_k(z) = \varphi_{k-1}(z) - \frac{1}{(k-1)!}. \tag{46}$$

See e.g. Hochbruck and Ostermann [15]. This property will be used throughout the remainder of the section.

**Lemma B.1.** *The transition matrices $\Phi(h), Q(h)$ of the once integrated Ornstein–Uhlenbeck process with rate parameter $L$ satisfy*

$$H\Phi(h) = [-L \quad I], \tag{47}$$

$$Q(h)H^\top = \begin{bmatrix} h^2 \varphi_2(Lh) \\ h\varphi_1(Lh) \end{bmatrix}, \tag{48}$$

$$HQ(h)H^\top = hI, \tag{49}$$

*Proof.*

$$H\Phi(h) = (E_1 - LE_0) \begin{bmatrix} I & h\varphi_1(Lh) \\ 0 & \varphi_0(Lh) \end{bmatrix} = [0 \quad \varphi_0(Lh)] - L[I \quad h\varphi_1(Lh)] = [-L \quad I]. \tag{50}$$

$$Q(h)H^\top = \int_0^h \begin{bmatrix} \tau^2 \varphi_1(L\tau)\varphi_1(L\tau)^\top & \tau\varphi_1(L\tau)\varphi_0(L\tau)^\top \\ \tau\varphi_0(L\tau)\varphi_1(L\tau)^\top & \varphi_0(L\tau)\varphi_0(L\tau)^\top \end{bmatrix} H^\top \, \mathrm{d}\tau \tag{51}$$

$$= \int_0^h \begin{bmatrix} \tau\varphi_1(L\tau)\varphi_0(L\tau)^\top - L\tau^2 \varphi_1(L\tau)\varphi_1(L\tau)^\top \\ \varphi_0(L\tau)\varphi_0(L\tau)^\top - L\tau\varphi_0(L\tau)\varphi_1(L\tau)^\top \end{bmatrix} \mathrm{d}\tau \tag{52}$$

$$= \int_0^h \begin{bmatrix} \tau\varphi_1(L\tau)\left(\varphi_0(L\tau)^\top - L\tau\varphi_1(L\tau)^\top\right) \\ \varphi_0(L\tau)\left(\varphi_0(L\tau)^\top - L\tau\varphi_1(L\tau)^\top\right) \end{bmatrix} \mathrm{d}\tau \tag{53}$$

$$= \int_0^h \begin{bmatrix} \tau\varphi_1(L\tau) \\ \varphi_0(L\tau) \end{bmatrix} \mathrm{d}\tau \tag{54}$$

$$= \begin{bmatrix} h^2 \varphi_2(Lh) \\ h\varphi_1(Lh) \end{bmatrix} \tag{55}$$

where we used $L\tau\varphi_1(L\tau) = \varphi_0(L\tau) - I$, and $\partial_\tau\left[\tau^k\varphi_k(L\tau)\right] = \tau^{k-1}\varphi_{k-1}(L\tau)$. It follows that

$$HQ(h)H^\top = H\begin{bmatrix} h^2\varphi_2(Lh) \\ h\varphi_1(Lh) \end{bmatrix} = h\left(\varphi_1(Lh) - Lh\varphi_2(Lh)\right) = hI, \tag{56}$$

where we used $L\tau\varphi_2(L\tau) = \varphi_1(L\tau) - I$. $\qquad\square$

**Lemma B.2.** *The prediction covariance $\Sigma_{n+1}^-$ satisfies*

$$\Sigma_{n+1}^-H^\top = Q(h)H^\top. \tag{57}$$

*Proof.* First, since the observation model is noiseless, the filtering covariance $\Sigma_n$ satisfies

$$H\Sigma_n = \begin{bmatrix} 0 & 0 \end{bmatrix}. \tag{58}$$

This can be shown directly from the correction step formula:

$$\begin{align}
H\Sigma_n &= H\Sigma_n^- - HK_nS_nK_n^\top \tag{59} \\
&= H\Sigma_n^- - H\left(\Sigma_n^-H^\top S_n^{-1}\right)S_nK_n^\top \tag{60} \\
&= H\Sigma_n^- - H\Sigma_n^-H^\top\left(H\Sigma_n^-H^\top\right)^{-1}S_nK_n^\top \tag{61} \\
&= H\Sigma_n^- - IS_nK_n^\top \tag{62} \\
&= H\Sigma_n^- - S_n\left(\Sigma_n^-H^\top S_n^{-1}\right)^\top \tag{63} \\
&= H\Sigma_n^- - S_nS_n^{-1}H\Sigma_n^- \tag{64} \\
&= \begin{bmatrix} 0 & 0 \end{bmatrix}. \tag{65}
\end{align}$$

Next, since the observation matrix is $H = \begin{bmatrix} -L & I \end{bmatrix}$, the filtering covariance $\Sigma_n$ is structured as

$$\Sigma_n = \begin{bmatrix} I \\ L \end{bmatrix}[\Sigma_n]_{00}\begin{bmatrix} I & L^\top \end{bmatrix}. \tag{66}$$

This can be shown directly from Eq. (58):

$$\begin{bmatrix} 0 & 0 \end{bmatrix} = H\Sigma = \begin{bmatrix} -L & I \end{bmatrix}\begin{bmatrix} \Sigma_{00} & \Sigma_{01} \\ \Sigma_{10} & \Sigma_{11} \end{bmatrix} = \begin{bmatrix} \Sigma_{10} - L\Sigma_{00} & \Sigma_{11} - L\Sigma_{01} \end{bmatrix}, \tag{67}$$

and thus

$$\begin{align}
\Sigma_{10} &= L\Sigma_{00}, \tag{68} \\
\Sigma_{11} &= L\Sigma_{01} = L\Sigma_{10}^\top = L\Sigma_{00}L^\top. \tag{69}
\end{align}$$

It follows

$$\Sigma = \begin{bmatrix} \Sigma_{00} & L\Sigma_{00} \\ \Sigma_{00}L^\top & L\Sigma_{00}L^\top \end{bmatrix} = \begin{bmatrix} I \\ L \end{bmatrix}\Sigma_{00}\begin{bmatrix} I & L^\top \end{bmatrix}. \tag{70}$$

Finally, together with Lemma B.1 we can derive the result:

$$\begin{align}
\Sigma_{n+1}^-H^\top &= \Phi(h)\Sigma_n\Phi(h)^\top H^\top + Q(h)H^\top \tag{71} \\
&= \Phi(h)\begin{bmatrix} I \\ L \end{bmatrix}\bar{\Sigma}_n\begin{bmatrix} I & L^\top \end{bmatrix}\begin{bmatrix} -L^\top \\ I \end{bmatrix} + Q(h)H^\top \tag{72} \\
&= \Phi(h)\begin{bmatrix} I \\ L \end{bmatrix}\bar{\Sigma}_n \cdot 0 + Q(h)H^\top \tag{73} \\
&= Q(h)H^\top. \tag{74}
\end{align}$$

$\qquad\square$

### B.3 Proof of Proposition 2

With these results, we can now prove Proposition 2.

*Proof of Proposition 2.* We prove the proposition by induction, showing that the filtering means are all of the form

$$\mu_n := \begin{bmatrix} y_n \\ Ly_n + N(\tilde{y}_n) \end{bmatrix}, \tag{75}$$

where $y_n, \tilde{y}_n$ are defined as

$$\tilde{y}_0 := y_0, \tag{76}$$
$$\tilde{y}_{n+1} := \varphi_0(Lh)y_n + h\varphi_1(Lh)N(\tilde{y}_n), \tag{77}$$
$$y_{n+1} := \varphi_0(Lh)y_n + h\varphi_1(Lh)N(\tilde{y}_n) - h\varphi_2(Lh)\left(N(\tilde{y}_n) - N(\tilde{y}_{n+1})\right). \tag{78}$$

This result includes the statement of Proposition 2.

**Base case $n = 0$**  The initial distribution of the probabilistic solver is chosen as

$$\mu_0 = \begin{bmatrix} y_0 \\ Ly_0 + N(\tilde{y}_0) \end{bmatrix}, \Sigma_0 = 0. \tag{79}$$

This proves the base case $n = 0$.

**Induction step $n \to n + 1$**  Now, let

$$\mu_n = \begin{bmatrix} y_n \\ Ly_n + N(\tilde{y}_n) \end{bmatrix} \tag{80}$$

be the filtering mean at step $n$ and $\Sigma_n$ be the filtering covariance. The prediction mean is of the form

$$\mu_{n+1}^- = \Phi(h)\mu_n = \begin{bmatrix} y_n + h\varphi_1(Lh)(Ly_n + N(\tilde{y}_n)) \\ \varphi_0(Lh)(Ly_n + N(\tilde{y}_n)) \end{bmatrix} = \begin{bmatrix} \varphi_0(Lh)y_n + h\varphi_1(Lh)N(\tilde{y}_n) \\ \varphi_0(Lh)(Ly_n + N(\tilde{y}_n)) \end{bmatrix}. \tag{81}$$

The residual $\hat{z}_{n+1}$ is then of the form

$$\hat{z}_{n+1} = E_1\mu_{n+1}^- - f(E_0\mu_{n+1}^-) \tag{82}$$
$$= \varphi_0(Lh)(Ly_n + N(\tilde{y}_n)) - f\left(\varphi_0(Lh)y_n + h\varphi_1(Lh)N(\tilde{y}_n)\right) \tag{83}$$
$$= \varphi_0(Lh)(Ly_n + N(\tilde{y}_n)) - L\left(\varphi_0(Lh)y_n + h\varphi_1(Lh)N(\tilde{y}_n)\right) - N\left(\tilde{y}_{n+1}\right) \tag{84}$$
$$= \varphi_0(Lh)Ly_n + \varphi_0(Lh)N(\tilde{y}_n) - L\varphi_0(Lh)y_n - Lh\varphi_1(Lh)N(\tilde{y}_n) - N\left(\tilde{y}_{n+1}\right) \tag{85}$$
$$= (\varphi_0(Lh) - Lh\varphi_1(Lh))N(\tilde{y}_n) - N\left(\tilde{y}_{n+1}\right) \tag{86}$$
$$= N(\tilde{y}_n) - N\left(\tilde{y}_{n+1}\right), \tag{87}$$
$$\tag{88}$$

where we used properties of the $\varphi$-functions, namely $Lh\varphi_1(Lh) = \varphi_0(Lh)$ and the commutativity $\varphi_0(Lh)L = L\varphi_0(Lh)$. With Lemma B.2, the residual covariance $S_{n+1}$ and Kalman gain $K_{n+1}$ are then of the form

$$S_{n+1} = H\Sigma_{n+1}^- H^\top = HQ(h)H^\top = hI, \tag{89}$$

$$K_{n+1} = \Sigma_{n+1}^- H^\top S_{n+1}^{-1} = Q(h)H^\top (hI)^{-1} = \begin{bmatrix} h\varphi_2(Lh) \\ \varphi_1(Lh) \end{bmatrix}. \tag{90}$$

This gives the updated mean

$$\mu_{n+1} = \mu_{n+1}^- - K_{n+1}\hat{z}_{n+1} \tag{91}$$
$$= \begin{bmatrix} \varphi_0(Lh)y_n + h\varphi_1(Lh)N(\tilde{y}_n) \\ \varphi_0(Lh)(Ly_n + N(\tilde{y}_n)) \end{bmatrix} - \begin{bmatrix} h\varphi_2(Lh) \\ \varphi_1(Lh) \end{bmatrix}(N(\tilde{y}_n) - N(\tilde{y}_{n+1})) \tag{92}$$
$$= \begin{bmatrix} \varphi_0(Lh)y_n + h\varphi_1(Lh)N(\tilde{y}_n) - h\varphi_2(Lh)\left(N(\tilde{y}_n) - N(\tilde{y}_{n+1})\right) \\ \varphi_0(Lh)(Ly_n + N(\tilde{y}_n)) - \varphi_1(Lh)\left(N(\tilde{y}_n) - N(\tilde{y}_{n+1})\right) \end{bmatrix}. \tag{93}$$

This proves the first half of the mean recursion:

$$E_0\mu_{n+1} = \varphi_0(Lh)y_n + h\varphi_1(Lh)N(\tilde{y}_n) - h\varphi_2(Lh)\left(N(\tilde{y}_n) - N(\tilde{y}_{n+1})\right) = y_{n+1}. \qquad (94)$$

It is left to show that

$$E_1\mu_{n+1} = Ly_{n+1} - N(\tilde{y}_{n+1}). \qquad (95)$$

Starting from the right-hand side, we have

$$Ly_{n+1} + N(\tilde{y}_{n+1}) \qquad (96)$$
$$= L\left(\varphi_0(Lh)y_n + h\varphi_1(Lh)N(\tilde{y}_n) - h\varphi_2(Lh)\left(N(\tilde{y}_n) - N(\tilde{y}_{n+1})\right)\right) + N(\tilde{y}_{n+1}) \qquad (97)$$
$$= \varphi_0(Lh)Ly_n + Lh\varphi_1(Lh)N(\tilde{y}_n) - Lh\varphi_2(Lh)\left(N(\tilde{y}_n) - N(\tilde{y}_{n+1})\right)N(\tilde{y}_{n+1}) \qquad (98)$$
$$= \varphi_0(Lh)Ly_n + (\varphi_0(Lh) - I)N(\tilde{y}_n) - (\varphi_1(Lh) - I)\left(N(\tilde{y}_n) - N(\tilde{y}_{n+1})\right)N(\tilde{y}_{n+1}) \qquad (99)$$
$$= \varphi_0(Lh)(Ly_n + N(\tilde{y}_n)) - \varphi_1(Lh)(N(\tilde{y}_n) - N(\tilde{y}_{n+1})) \qquad (100)$$
$$= E_1\mu_{n+1}. \qquad (101)$$

This concludes the proof of the mean recursion and thus shows the equivalence of the two recursions.

$\square$

## C  Proof of Proposition 3: L-stability

We first provide definitions of L-stability and A-stability, following [27, Section 8.6].

**Definition 1** (L-stability). *A one-step method is said to be L-stable if it is A-stable and, in addition, when applied to the scalar test-equation $\dot{y}(t) = \lambda y(t)$, $\lambda \in \mathbb{C}$ a complex constant with $\mathrm{Re}(\lambda) < 0$, it yields $y_{n+1} = R(h\lambda)y_n$, and $R(h\lambda) \to 0$ as $\mathrm{Re}(h\lambda) \to -\infty$.*

**Definition 2** (A-stability). *A one-step method is said to be A-stable if its region of absolute stability contains the whole of the left complex half-plane. That is, when applied to the scalar test-equation $\dot{y}(t) = \lambda y(t)$ with $\lambda \in \mathbb{C}$ a complex constant with $\mathrm{Re}(\lambda) < 0$, the method yields $y_{n+1} = R(h\lambda)y_n$, and $\{z \in \mathbb{C} : \mathrm{Re}(z) < 0\} \subset \{z \in \mathbb{C} : R(z) < 1\}$.*

*Proof of Proposition 3.* Both L-stability and A-stability directly follow from Remark 1: Since the probabilistic exponential integrator solves linear ODEs exactly its stability function is the exponential function, i.e. $R(z) = \exp(z)$. A-stability and L-stability then follow: Since $\mathbb{C}^- \subset \{z : |R(z)| \leq 1\}$ holds the method is A-stable. And since $|R(z)| \to 0$ as $\mathrm{Re}(z) \to -\infty$ the method is L-stable. $\square$

## D  Experiment details

### D.1  Burger's equation

Burger's equation is a semi-linear partial differential equation (PDE) of the form

$$\partial_t u(x, t) = -u(x, t)\partial_x u(x, t) + D\partial_x^2 u(x, t), \qquad x \in \Omega, \quad t \in [0, T], \qquad (102)$$

with diffusion coefficient $D \in \mathbb{R}_+$. We discretize the spatial domain $\Omega$ on a finite grid and approximate the spatial derivatives with finite differences to obtain a semi-linear ODE of the form

$$\dot{y}(t) = D \cdot L \cdot y(t) + F(y(t)), \qquad t \in [0, T], \qquad (103)$$

with $N$-dimensional $y(t) \in \mathbb{R}^N$, $L \in \mathbb{R}^{N \times N}$ the finite difference approximation of the Laplace operator $\partial_x^2$, and a non-linear part $F$.

More specifically, we consider a domain $\Omega = (0, 1)$, which we discretize with a grid of $N = 250$ equidistant locations, thus we have $\Delta x = 1/N$. We consider zero-Dirichlet boundary conditions, that is, $u(0, t) = u(1, t) = 0$. The discrete Laplacian is then

$$[L]_{ij} = \frac{1}{\Delta x^2} \cdot \begin{cases} -2 & \text{if } i = j, \\ 1 & \text{if } i = j \pm 1, \\ 0 & \text{otherwise.} \end{cases} \qquad (104)$$

The non-linear part of the discretized Burger's equation results from another finite-difference approximation of the term $u \cdot \partial_x u$, and is chosen as

$$[F(y)]_i = \frac{1}{4\Delta x} \begin{cases} y_2^2 & \text{if } i = 1, \\ y_{d-1}^2 & \text{if } i = d, \\ y_{i+1}^2 - y_{i-1}^2 & \text{else.} \end{cases} \tag{105}$$

The initial condition is chosen as

$$u(x, 0) = \sin(3\pi x)^3 (1 - x)^{3/2}. \tag{106}$$

We consider an integration time-span $t \in [0, 1]$, and choose a diffusion coefficient $D = 0.075$.

## D.2 Reaction-diffusion model

The reaction-diffusion model presented in the paper, with logistic reaction term, has been used to describe the growth and spread of biological populations [22]. It is given by a semi-linear PDE

$$\partial_t u(x, t) = D \partial_x^2 u(x, t) + R(u(x, t)), \qquad x \in \Omega, \quad t \in [0, T], \tag{107}$$

where $D \in \mathbb{R}_+$ is the diffusion coefficient and $R(u) = u(1 - u)$ is a logistic reaction term. We discretize the spatial domain $\Omega$ on a finite grid and approximate the spatial derivatives with finite differences, and obtain a semi-linear ODE of the form

$$\dot{y}(t) = D \cdot L \cdot y(t) + R(y(t)), \qquad t \in [0, T], \tag{108}$$

with $N$-dimensional $y(t) \in \mathbb{R}^N$, $L \in \mathbb{R}^{N \times N}$ the finite difference approximation of the Laplace operator, and the reaction term $R$ is as before but applied element-wise.

We again consider a domain $\Omega = (0, 1)$, which we discretize on a grid of $N = 100$ points. This time we consider zero-Neumann conditions, that is, $\partial_x u(0, t) = \partial_x u(1, t) = 0$. Including these directly into the finite-difference discretization, the discrete Laplacian is then

$$[L]_{ij} = \frac{1}{\Delta x^2} \cdot \begin{cases} -1 & \text{if } i = j = 1 \text{ or } i = j = d, \\ -2 & \text{if } i = j, \\ 1 & \text{if } i = j \pm 1, \\ 0 & \text{otherwise.} \end{cases} \tag{109}$$

The initial condition is chosen as

$$u(x, 0) = \frac{1}{1 + e^{30x - 10}}. \tag{110}$$

The discrete ODE is then solved on a time-span $t \in [0, 2]$, and we choose a diffusion coefficient $D = 0.25$.

