# OpenReview forum: "Probabilistic Exponential Integrators"
_NeurIPS.cc/2023/Conference — NeurIPS 2023 poster_

### Official Review · Reviewer_Lu6G · 2023-07-01

**Soundness:** 4 excellent
**Presentation:** 4 excellent
**Contribution:** 3 good
**Rating:** 8
**Confidence:** 4

**Summary:**

This paper proposes a probabilistic integrator for solving stiff semi-linear ODEs that decouples linear and nonlinear components for improved accuracy.  The approach uses an integrated Ornstein-Uhlenbeck process (IOUP) prior and posterior inference extends classical exponential integrator to exactly solve the linear components.  Nonlinearities are approximated via local linearization similar to the extended Kalman filter (EKF).  The authors present L-stability results and address computation through proposed approximations of the Jacobian that capture only the linear dynamics.

**Strengths:**

This is a solid paper; well-written, well-motivated, nice execution.  The results aren’t groundbreaking when compared to baseline, but there are clear benefits of the proposed method, particularly when nonlinearities are less significant.  Furthermore, results are presented honestly and limitations are clearly identified.

**Weaknesses:**

The paper fails to highlight any benefit of treating the initial value problem (IVP) as a Bayesian inference task.  The posterior distribution *should* characterize uncertainty over the solution, and thus it is possible to quantify and report confidence in the solution.  It is further possible to scrutinize the appropriateness of the IOUP as a prior.  The paper does not address this, but instead prioritizes numerical accuracy and execution time.  While these metrics are certainly important, they are not the primary motivation of probabilistic numerics.  This is not a critical flaw, but perhaps a missed opportunity to demonstrate the benefit of a Bayesian approach to a numerical method.

Unsurprisingly, the proposed method is fairly sensitive to nonlinearity of the ODEs.  The experiment in Sec. 4.1 demonstrates that accuracy degrades as the quadratic term becomes more significant.  In fact, the proposed method seems to be more sensitive to this than baseline IWP methods, albeit with uniformly lower error.  For the simple ODE in Sec. 4.1 all methods achieve very low error (on the order of $10^{-10}$) so the observed improvement with the IOUP is marginal unless the ODE is highly linear.

Some detailed comments:
  * Fig. 5 should reference subfigures (e.g. (a) is never explicitly mentioned).
  * L268: Typo "Since th problem..."

**Questions:**

* The authors suggest that probabilistic integrators may be preferable for expensive-to-evaluate functions (L259).  How?  Can the method avoid explicit evaluation via prediction?

* The EKF can be quite unreliable in even simple nonlinear dynamical systems, leading to preferred approaches such as the UKF.  Would not such an approach be preferable here to avoid the local linear approximations?  What is the main challenge in a more accurate approximation?

**Limitations:**

Limitations are clearly described in Sec. 5

---

> ### Author Rebuttal · Authors · 2023-08-03
>
> We first want to thank the reviewer for their insightful comments, feedback, and
> questions. In the following we will address each point separately.
>
>
> > The paper fails to highlight any benefit of treating the initial value problem
> (IVP) as a Bayesian inference task. The posterior distribution should
> characterize uncertainty over the solution, and thus it is possible to quantify
> and report confidence in the solution. It is further possible to scrutinize the
> appropriateness of the IOUP as a prior. The paper does not address this, but
> instead prioritizes numerical accuracy and execution time. While these metrics
> are certainly important, they are not the primary motivation of probabilistic
> numerics. This is not a critical flaw, but perhaps a missed opportunity to
> demonstrate the benefit of a Bayesian approach to a numerical method.
>
>
> The reviewer brings up a fair point. However, we would like to point out the
> paucity of papers critically examining different priors. In fact, to our
> knowledge, this is the first to give a clear argument for using anything other
> than the canonical IWP prior. With this in mind, we hope the present
> contribution can spark interest in taking the prior selection more seriously in
> upcoming investigations by the PN community.
>
>
> > Unsurprisingly, the proposed method is fairly sensitive to nonlinearity of the
> ODEs. The experiment in Sec. 4.1 demonstrates that accuracy degrades as the
> quadratic term becomes more significant. In fact, the proposed method seems to
> be more sensitive to this than baseline IWP methods, albeit with uniformly lower
> error.
>
> The performance benefits of the probabilistic exponential integrator with IOUP
> prior over a solver with IWP prior depend compeletely on the known linear part
> opf the ODE. If the linear part is set to zero, they both coincide and the
> IOUP-based solver performs exactly as the solver with IWP prior. This behaviour
> is exactly what the experiment in Section 4.1 and Figure 3 aim to visualize,
> showing that as the linear part increasingly dominates the dynamics, the benefit
> of using an IOUP prior increases. Note also that in this experiment the IOUP
> prior never performs worse than the EKL&IWP combination, so we see this not as
> "sensitivity to nonlinearity" but rather as "leveraging the semi-linearity".
>
>
> > The authors suggest that probabilistic integrators may be preferable for
> expensive-to-evaluate functions (L259). How? Can the method avoid explicit
> evaluation via prediction?
>
> The comment in line 259 refers to choosing between the standard and Rosenbrock
> version of the proposed probabilistic exponential integrator, and the suggestion
> stems purely from the required number of function evaluations per step: Since
> the local linearization of the Rosenbrock method requires evaluating the vector
> field, the standard probabilistic exponential integrator requires less
> evaluations and thus might be preferrable when the vector field becomes
> exceedingly expensive to evaluate. But note that both methods perform at least
> one evaluation of the vector field per step.
>
>
> > The EKF can be quite unreliable in even simple nonlinear dynamical systems,
> leading to preferred approaches such as the UKF. Would not such an approach be
> preferable here to avoid the local linear approximations? What is the main
> challenge in a more accurate approximation?
>
> This is a very good point. It is indeed well known that the UKF or other
> filtering/smoothing methods can be advantageous over the EKF/EKS in certain
> state estimation problems. In the specific context of probabilistic ODE solvers,
> the UKF has been previously suggested [1,2], but a more extensive evaluation of
> its properties and utilities would certainly be interesting in order to better
> understand for which types of problems it would be most beneficial.
>
> Note also that in the context of probabilistic ODE solvers, a more accurate
> linearization not the only way to improve the accuracy of the method: we could
> also just select a smaller step size. This is not the case for most standard
> filtering problems. And in the regime of "very small" steps, a local Taylor
> approximation becomes increasingly accurate.
>
>
> [1] Kersting et al, "Active uncertainty calibration in Bayesian ODE solvers",
> UAI (2016)
>
> [2] Tronarp et al, "Probabilistic solutions to ordinary differential equations
> as nonlinear Bayesian filtering: a new perspective", Statistics and Computing
> (2019)

---

> > ### Comment · Reviewer_Lu6G · 2023-08-11
> >
> > Thanks.  Your responses cleared up a couple points.  I will keep my scores as-is and am willing to argue in favor of the paper.

---

### Official Review · Reviewer_V9iy · 2023-07-06

**Soundness:** 3 good
**Presentation:** 3 good
**Contribution:** 3 good
**Rating:** 7
**Confidence:** 1

**Summary:**

This paper presents probabilistic exponential integrators as a solver for stiff semi-linear ODEs.Their solver can also be extended to solve general non-linear ODEs with iterative re-linearization. The proposed method is shown to be L-stable in theory and empirically more stable than existing methods.

**Strengths:**

The paper is well written. It tackles the problem of solving stiff systems, which is a great challenge in probabilistic numerics. The authors exploit the properties in semi-linear ODEs and develop a probabilistic solver that integrates the fast, linear dynamics into the prior model. The solver is also extended to general nonlinear ODEs via iterative linearization. Both theoretical and empirical analysis are sound.

**Weaknesses:**

The main difficulty of conventional of stiff ODEs solvers is its sensitivity to step size: small step size is required for stability. Proposition 3 shows the proposed solver is L-stable. However, the experiments mainly focus on final errors across various step sizes. A detailed investigation of stability (boundedness of solutions?) behaviors of solvers under large step size may help readers better understand the proposed approach.

**Questions:**

See Weakness

**Limitations:**

See Weakness

---

> ### Author Rebuttal · Authors · 2023-08-03
>
> We first want to thank the reviewer for their insightful comments and feedback.
>
> > the experiments mainly focus on final errors across various step
> sizes. A detailed investigation of stability (boundedness of solutions?)
> behaviors of solvers under large step size may help readers better understand
> the proposed approach.
>
> Thank you for bringing this up. The main advantage of probabilistic exponential
> integrators as introduced in the paper is indeed their improved stability. A
> more detailed investigation purely focused on stability could have been helpful
> to make this message more clear; but we want to highlight that the improved
> performance for large step sizes is demonstrated in the experiments: In both
> Figure 4 and 5, the "error vs. step size" plots show that as step sizes
> increase, the non-exponential methods start to fail (either by diverging or
> achieving very large errors >>1) whereas the exponential methods still provide
> meaningful solutions; see for example step size $10^{-1}$ in both figures. This
> demonstrates the improved stability of the exponential integrators.

---

> > ### Comment · Reviewer_V9iy · 2023-08-21
> >
> > Thanks for for response. I will keep my score

---

### Official Review · Reviewer_vChc · 2023-07-17

**Soundness:** 3 good
**Presentation:** 3 good
**Contribution:** 3 good
**Rating:** 6
**Confidence:** 3

**Summary:**

This paper proposed probabilistic exponential integrators, which is a new class of probabilistic solvers for stiff semi-linear ODEs. More specifically, the integrated OU process is introduced for a functional prior that directly incorporate the linear part of the dynamics. As a result, the proposed methods can be more stable than their non-exponential probabilistic counterparts. In addition, the authors also provide an extension for general non-linear systems, which can be viewed as a probablistic solver based on exponential Rosenbrock-type methods.

**Strengths:**

1. The paper is overall clearly written and nicely organized.
2. The IOUP prior is neat. Also, some theoretical results of the proposed probabilistic exponential integrator have been established (e.g., equivalence to the classic exponential trapezoidal rule and L-stability).

**Weaknesses:**

1. The proposed methods are only for semi-linear ODEs, while other baselines can be applied to more general cases.
2. The IOUP prior is more expensive than the IWP prior.
3. The Kalman filter/smoother based solver has cubic scaling in the ODE dimension.
4. It seems that the advantage of the method is more significant when the step size is large (error is large), and becomes negligible when the step size is small.

Note that 2 and 3 have been adimitted by the authors.

**Questions:**

1. What is the overhead computational time for computing the matrix exponentials? Is it contained in the runtime?
2. The proposed method can be more accurate when the linear part is more dominant. Since EK1 also introduces a linear approximation, would it perform better when combined with the IOUP prior?

**Limitations:**

Yes, they do.

---

> ### Author Rebuttal · Authors · 2023-08-03
>
> We first want to thank the reviewer for their comments, feedback, and questions.
> In the following we will address each open point separately.
>
> > Weakness 1. The proposed methods are only for semi-linear ODEs, while other baselines can be applied to more general cases.
>
> The main "probabilistic exponential integrator" proposed in the paper is
> indeed only fully applicable to semi-linear ODEs. But, as also stated
> in the Summary section of your review, the proposed Rosenbrock-type
> method of Section 3.6 is applicable to general non-linear ODEs.
>
> > Weakness 4. It seems that the advantage of the method is more significant when the step size is large (error is large), and becomes negligible when the step size is small.
>
> This is a very good point, and this is precisely why "stability" is so important for numerical ODE solvers. If the step size is "small enough", even the simplest forward Euler method eventually computes an accurate solution---but the problem is that for extremely small steps the runtime becomes prohibitively large. By using more stable solvers, such as exponential integrators, accurate solutions can be computed with larger step sizes.
>
> > Question 1. What is the overhead computational time for computing the matrix exponentials? Is it contained in the runtime?
>
> The cost of computing the matrix exponential depends on the drift matrix and its
> size, and thus on the specific ODE and the order of the prior. For fixed step
> sizes, the proposed probabilistic exponential integrator computes _one_ matrix
> exponential, and then re-uses the computed transition matrices for each step.
> Thus, it computes exaclty one matrix exponential more than a solver with IWP
> prior. The Rosenbrock-type method however needs to compute a matrix exponential
> _at each timestep_ as the linear part changes at each step, and is thus
> computationally more demanding, but also more stable, as shown in Figured 4
> and 5. The reported runtimes include all parts of the solver and thus also the
> computation of the matrix exponential.
>
>
> > Question 2. The proposed method can be more accurate when the linear part is more dominant. Since EK1 also introduces a linear approximation, would it perform better when combined with the IOUP prior?
>
> This is a very good question. Indeed, the proposed method can also be combined
> with the exact first-order Taylor linearization (EK1). To keep the number of
> methods in the experiments and figures reasonably small, we have chosen to
> combine the probabilistic exponential integrator only with the EKL (thus not
> requiring any additional linearization during the solve), and the
> Rosenbrock-type method with the EK1 (as both the Rosenbrock-type method and the
> EK1 require local linearizations anyways); but both solvers can be combined with
> the different linearization strategies mentioned in Section 3.3. In experiments
> not included in the paper, we have also tested the non-Rosenbrock method with
> the EK1, and we have observed that this can be more accurate than the EKL---but
> with the additional cost/downside of requiring local linearizations.

---

> > ### Comment · Reviewer_vChc · 2023-08-16
> > **Thanks for the reply**
> >
> > Thanks for the response. I will keep my score.

---

### Official Review · Reviewer_goRD · 2023-07-24

**Soundness:** 2 fair
**Presentation:** 2 fair
**Contribution:** 2 fair
**Rating:** 5
**Confidence:** 4

**Summary:**

The paper studies classical exponential integrators using the framework of probabilistic numerics, where Bayesian formalism is used to study numerical approximations of deterministic dynamical systems. The main result seems to be Proposition 1, which establishes exponential stability (called L-Stability for some reason) property for the exponential integrator.

**Strengths:**

The paper reasonably well introduces probabilistic numerics and studies this in the context of ODEs and tackles important notion of stability of numerical approximation.

**Weaknesses:**

The paper transfers well-understood properties of numerical approximations of ODEs using the formalism of probabilistic numerics. In my view, this is not a sufficiently novel contribution.

**Questions:**

- Calling the Ornstein-Uhlenbeck process a Gauss-Markov in eqn 3 is overkill.
- Notation in eqn 5 of normal distribution not clear what. Why three parameters?
- Section 2.2 I don't see why \mathcal I [y] = 0  when y is a solution to ODE. This says that y_t-f(y_t,t)=0 not \frac{d}{dt} y_t -f(y_t,t)=0
- Explain the role of Z in 9b
- Paper seems to treat ODEs, but some numerical examples are derived for PDEs. Of course, semi-group theory for PDEs links with exponential integrators, but this seems to go beyond this paper.

**Limitations:**

Methodology applies to a particular class of ODES

---

> ### Author Rebuttal · Authors · 2023-08-03
>
> We first want to thank the reviewer for their comments, feedback, and questions.
> In the following we will address each open point separately.
>
> > Calling the Ornstein-Uhlenbeck process a Gauss-Markov in eqn 3 is overkill.
>
> Probabilistic numerical ODE solvers have in general been established with any
> Gauss--Markov prior that allows for easy access of derivatives [1]. Since
> section 2 is the background section of the paper, we introduced ODE filters in
> their generality.
>
> > Notation in eqn 5 of normal distribution not clear what. Why three parameters?
>
> The first entry of `N(x; m, C)` is not a parameter, but the variable that is
> described by the Gaussian. This is just a slighly more thorough notation than
> the equivalent short-hand notation `N(m, C)`. To prevent confusion, we will make
> sure that the notation is consistent accross the paper.
>
> > Section 2.2 I don't see why \mathcal I [y] = 0 when y is a solution to ODE.
> This says that y_t-f(y_t,t)=0 not \frac{d}{dt} y_t -f(y_t,t)=0
>
> $E_1$ is a selection matrix that selects the entry that corresponds to the first
> derivative, and similarly $E_0$ selects the zeroth derivative. In terms of the
> Gauss--Markov state representation $Y$, $\mathcal{I}$ therefore encodes exactly
> the ODE. The statement $\mathcal{I}\[y\] \equiv 0$ comes with a slight abuse of
> notation. We could also define $\mathcal{I}$ not via the "state" $Y$ as in eq.
> (7), but in the space of the $d$-dimensional function $y$, as
> $\mathcal{I}\[y\](t) = D y(t) - f(y(t), t)$,
> where $D$ is the derivative operator. On $Y^{(0)}$ this then gives
> $\mathcal{I}\[Y^{(0)\}](t) = D Y^{(0)}(t) - f(Y^{(0)}(t), t) = Y^{(1)}(t) - f(Y^{(0)}(t), t)$.
> This formula is equivalent to the formulation in the paper (eq. 7) and, since
> the algorithm only operates on the state $Y$, it also closer to the actual
> implementation of the method.
>
> > Explain the role of Z in 9b
>
> $Z_n$ formally describe the data, _before_ it is actually observed. Then,
> when we add the actual data to the model as defined in eq. (9), we can do
> (approximate) Bayesian inference. This notation with prior and likelihood model
> is standard in Bayesian filtering and smoothing (and Bayesian inference in
> general). In the specific case of probabilistic ODE solvers the data is exactly
> zero everywhere, since we want the ODE to hold on the grid.
>
> > Paper seems to treat ODEs, but some numerical examples are derived for PDEs.
>
> The method of lines can be used to transform semi-linear PDEs into semi-linear
> ODEs, as we did in the experiment section. The semi-linear ODE is then solved
> with the proposed method. Note that spatial discretizations of PDEs are also a
> very common example to motivate exponential integrators; see for instance the
> introduction of [2].
>
> > Methodology applies to a particular class of ODES
>
> The goal of the paper was indeed to develop methods particularly for stiff
> semi-linear ODEs. And in the context of numerical ODE solvers, developing
> specific methods for specific problems is quite common. We still want to
> highlight Section 3.6 of our paper: The proposed exponential integrator can also
> be applied to any nonlinear ODE by automatically and continuously linearizing
> the problem, in a manner that is very similar to classic exponential Rosenbrock
> methods.
>
>
> [1] Tronarp et al, "Bayesian ODE solvers: the maximum a posteriori estimate",
> Statistics and Computing (2021)
>
> [2] Hochbruck et al, "Exponential Rosenbrock-Type Methods", SIAM Numerical
> Analysis (2009)

---

> > ### Comment · Reviewer_goRD · 2023-08-14
> >
> > Thank you for clarifying a few points and answering my questions. As a result of these, I increased my score by 1.

---

### Decision · Program_Chairs · 2023-09-21

**Decision:**

Accept (poster)

**Comment:**

This is a well-written contribution that narrows the performance gap between classical and probabilistic numerical methods for ODEs and I am delighted to agree with the Reviewers that it should be accepted.